# The Effectiveness of Healthcare System Resilience during the COVID-19 Pandemic: A Case Study

**DOI:** 10.3390/medicina59050946

**Published:** 2023-05-14

**Authors:** Monika Borzuchowska, Dorota Kilańska, Remigiusz Kozłowski, Petre Iltchev, Tomasz Czapla, Sylwia Marczewska, Michał Marczak

**Affiliations:** 1Department of Management and Logistics in Healthcare, Medical University of Lodz, 90-131 Lodz, Poland; monika.borzuchowska@stud.umed.lodz.pl (M.B.); remigiusz.kozlowski@umed.lodz.pl (R.K.);; 2Institute of Nursing and Midwifery, Medical University of Gdansk, 80-210 Gdansk, Poland; 3Department of Coordinated Care, Medical University of Lodz, Al. Kościuszki 4, 90-131 Lodz, Poland; 4Department of Management, Faculty of Management, University of Lodz, 90-237 Lodz, Poland; 5Collegium of Management WSB University of Warsaw, 03-204 Warsaw, Poland

**Keywords:** resilience, nurse, ICU, nursing care, healthcare system, COVID-19

## Abstract

*Introduction*: The outbreak of the COVID-19 pandemic was a period of uncertainty and stress for healthcare managers due to the lack of knowledge (about the transmission of the virus, etc.) and also due to the lack of uniform organisational and treatment procedures. It was a period where the ability to prepare for a crisis, to adapt to the existing conditions, and to draw conclusions from the situation were of critical importance to keep ICUs (intensive care units) operating. The aim of this project is to compare the pandemic response to COVID-19 in Poland during the first and second waves of the pandemic. This comparison will be used to identify the strengths and weaknesses of the response, including challenges presented to health professionals and health systems and ICUs with COVID-19 patients according to the European Union Resilience Model (2014) and the WHO Resilience Model (2020). The WHO Resilience model was suitable to the COVID-19 situation because it was developed based on this experience. *Methods*: A matrix of 6 elements and 13 standards assigned to them was created using the EC and WHO resilience guidelines. *Results*: Good governance in resilient systems ensures access to all resources without constraints, free and transparent flow of information, and a sufficient number of well-motivated human resources. *Conclusions*: Appropriate preparation, adaptation to the existing situation, and effective management of crisis situations are important elements of ensuring the resilience of ICUs.

## 1. Introduction

Resilience in health systems is a key ingredient of readiness for challenges such as catastrophic events and unexpected situations such as disasters and pandemics. Good governance requires, among other things, a transparent flow of information, which is also a key element of resilience necessary to ensure resilient management in health care. At a practical level, a manager requires stable access to an uninterrupted flow of all kinds of resources, should have the ability to react to various crises, and be able to manage in crisis situations. This should facilitate an appropriate response to such turmoil as the WHO’s ‘Severe Acute Respiratory Syndrome coronavirus 2’ (SARS-CoV-2) pandemic, later named COVID-19 [1].

The pandemic also had consequences for medical staff. During the SARS-CoV-2 outbreak, nurses experienced more stress than other hospital staff [2,3]. In the literature, the significant impact of multiple factors is emphasized, i.e., contact with death, stress, and professional burnout, which impacted nurses’ decisions to resign from the profession after working during the COVID-19 pandemic [4].

### Healthcare Systems Resilience—A Global Context

Ensuring resilience, which could constitute a remedy for the issue of health debt in a pandemic, is a significant challenge for healthcare systems. After the Ebola epidemic (in 2014) we gained knowledge of what to do and how to appropriately react to the next catastrophic crisis [1,5,6]. Both the World Health Organisation (WHO) [7] and the European Commission (EC) have provided management support tools [8].

Resilience entails access to Universal Health Coverage (UHC), understood as: 1. An adequate number of trained health workers; 2. Available medicines; 3. Robust health information systems, including surveillance; 4. Appropriate infrastructure; 5. Sufficient public financing; and 6. A strong public sector to deliver equitable, quality services [9]. Definitions of health system resilience are focused on understanding the preparedness of a health system and its ability to absorb, adapt, and transform in order to cope with acute shocks to health and the economy [10]. According to Thomas, ‘prepare for’ has an important meaning, recognizing that resilience has a forward-looking element that seeks to reduce the risks from the impact of future health system shocks, which can be understood as a sudden and extreme (severe) change which impacts a health system. The response of a health system to a shock can be seen as a cycle consisting of four stages. This article’s authors focused on Stage 1: preparedness is related to how vulnerable a system is to various disturbances (limiting exposure) and how ready it is for when a shock hits (e.g., by having practised and resourced systems of response) and the following definition of resilience according to WHO was adopted: 'resilience as the ability to prepare for, manage (absorb, adapt, and transform) and learn from shocks' [7].The European Commission specified the main pillars of resilience in 2014, at the same time requesting member states to prepare for another healthcare crisis. The specified elements of resilience include: stable funding mechanisms; sound risk adjustment methods; good governance; information flows in the system and adequate costing of health services; and a health workforce of adequate capacity and with the right skills [8]. They are all important and interdependent. It is believed that the process of creating resilience may be supported by improvements to healthcare policy, directing it towards the promotion of health and prevention of infectious diseases, and also by structural reforms which ensure maintaining the balance of healthcare financing [11].

## 2. Materials and Methods

### 2.1. Research Objectives

The aim of this project is to compare the pandemic response to COVID-19 in Poland during the first and second waves of the pandemic. This comparison will be used to identify the strengths and weaknesses of the response, including challenges presented to health professionals and health systems and ICUs with COVID-19 patients according to the European Union Resilience Model (2014) and the WHO Resilience Model (2020). The WHO Resilience model was suitable to the COVID-19 situation because it was developed based on this experience. The research questions are as follows:

Q1. How was the response planned, organized, and implemented in Poland?

Q2. How was the response planned, organized, and implemented in one ICU in Poland during COVID-19?

Q3. What impact did the number of nurses and their absences have on patient deaths?

Q4. What impact did the preparation of the health system have on the preparation of the intensive care unit to care for patients during the COVID-19 pandemic?

### 2.2. Research Design

In the field of health systems research, comparative approaches are recommended [12] and are essential in order to develop practical, transferable lessons. We used a case study approach with multiple levels of nested analysis [13]. The healthcare system and ICUs in hospitals and the public health interventions were considered as a single case, where the ICU situation is described in the context of the national health situation.

In the case of the entire system and the hospital case study (Q1 and Q2), the analysis corresponds to the different meanings of the different configurations. Configurations were identified using a comparative perspective based on the European Commission 2014 Resilience Model conceptual framework consisting of 6 elements. In order to reflect the situation in 2020, reference was made to the WHO Resilience Model [7], which was selected using the Delphi method. Experts were appointed from among the experts of the InterDoctorMen project. The team of experts consisted of 6 people: 3 were logistics and management experts from the University of Lodz and Medical University of Lodz (1), 2 experts were nurses in the field of care quality, management, and international nursing, and one expert was a statistician. The study was conducted by a PhD student. The elements of the WHO Resilience Model were assigned to selected elements of the EC 2014 Resilience Model using a consensus process, in such a way that at least one element of the WHO Resilience Model (a–i) was assigned to each of the elements of the EC 2014 Resilience Model.

Based on the consensus of experts, the resilience models describe the preparation of the healthcare system and of the ICU for the COVID-19 pandemic. The case study analysis of the system was conducted on the basis of publicly available statistical data, legal acts, and information from the Ministry of Health. The ICU description was based on the example of one hospital: Pirogow Hospital in Lodz, which provided data for the analysis of staff resources, nurse absenteeism, and patient compliance, and where the PhD student conducting the research was employed (Q3).

The method of observation and analysis of ICU documents conducted by the PhD student was a case study method. On this basis, an assessment of the ward’s preparation for the COVID-19 pandemic was carried out, using the elements of the EC Resilience Model (I–VI). It was assumed that in the case of the EC’s Good management (I), the following may be important: Effective and participatory leadership with a strong vision and communication (a) and Coordination of activities across government and key Stakeholders (b) [7]. Whereas for Information flows in the system (IV), the Effective information systems and flows (d) are important.

The descriptions of the cases of the system and the ICU were combined, demonstrating the preparation of the ICU against the background of the preparation of the healthcare system in Poland for the COVID-19 pandemic.

For the case studies in public health (Q4), we focused on understanding whether and how the system is prepared for SARS-CoV-2 according to the EC Resilience Model from 2014, supplemented with WHO 2020 variables. Each element of the model was treated as a case study description at the country level.

### 2.3. Data Collection

For Q1 and Q2, we describe how the system’s and hospital’s (ICU’s) responses to COVID-19 were planned, organized, and implemented in order to describe the resilience of hospitals and their staff. Several techniques for collecting empirical data were used (observation and analysis of documents). The result of the observations of the work of nurses in the ward is a description of the preparation of nurses to work during a pandemic.

For the purposes of Q3, an analysis of hospital data was conducted, selecting 3 months from the 2020–2022 period. Several empirical data collection techniques were used (observation, data collection, and document analysis). Due to the range of available hospital data, January 2020 (pre-pandemic), January 2021, and January 2022 (during the pandemic) were included in the comparative analyses.

For Q4, several empirical data collection techniques were used (observation, data collection, and document analysis).

## 3. Results

### Readiness of the System for a Pandemic-Caused Crisis

The team of authors conducted an analysis of the literature and a semantic analysis of the elements of the WHO strategy and EC pillars, creating a matrix (Table 1) which logically combined pillars and strategies.

*Good governance* [8] in resilience applies to the roles of organisational bodies at all levels of management (WHO) [7]. This is primarily an issue of cooperation between sectors, of agreements with, e.g., international agencies, service providers, and non-governmental organisations, etc. The Effective and participatory leadership with a strong vision and communication (a) and Coordination of activities across government and key Stakeholder (b) were assigned to this category, as well as Organizational learning culture that is responsive to crises (c).

In the scope of Coordination of activities across government and key Stakeholders, in which inter-sector cooperation is necessary [7], in March of 2020 only 32 out of 47 countries 'managed the pandemic' on a governmental level. Due to the preparation (competencies) of the nurses and the available equipment, the ICUs were key locations where services were provided to COVID-19 patients. The organisation of an ICU, the necessary equipment, and the detailed requirements concerning the qualification of medical staff were standardised [7]. Due to their inefficiency, 21 hospitals in Poland were transformed into COVID-19 hospitals [14], and in every province a temporary hospital was created, thus increasing the number of beds with the necessary infrastructure by 5500. At the central level, the National Consultant in Nursing has established procedures [15] based on the guidelines issued by the Ministry of Health [16,17,18].

IGood governance understood as effective and participatory leadership with a strong vision and communication [7,8] and strong and flexible leadership [19] has a special impact on employees during a crisis. A strong connection/link between manager resilience and employees’ problem-solving ability was demonstrated [20,21]. The manager’s responsibilities included: adapting the infrastructure to existing needs, managing available competencies, monitoring risks, following procedures, and securing personal protective equipment (PPE) (PPE are means of protecting face and eyes, the airways, lower limbs, or body) [22,23]. In 2021, 34% of all countries reported a lack of PPE, a lack of the possibility of financing PPE, 22% reported a lack of distribution capacities, and 1 out of 10 WHO countries stated a lack of clear procedures for the handling of PPE [24]. The report has no data from Poland.Effective and participatory leadership with a strong vision and communication (a) [7] is understood by the authors to mean, among others, good international cooperation, and sharing of procedural models or of data necessary for the creation of predictive models. We focused here on the communication of data, using it to create databases and ensuring the flow of information necessary to make administrative and clinical decisions [1]. Nursing care data were unavailable. In many countries, the recommended standardised nursing terminology [25] was not implemented in nursing practice [26]. In Poland, it was piloted in electronic documentation, and the results of the pilot programme were published in December of 2020. There were no tools available for the assessment of the impact of nursing care on, e.g., patient deaths [26,27]. The nurses did not prepare and manage care plans (a nursing care plan (NCP) is a formal process that correctly identifies existing needs and recognizes potential needs or risks; nursing care plan formats are usually categorized or organized into four columns: (1) nursing diagnoses, (2) desired outcomes and goals, (3) nursing interventions, and (4) evaluations), and they had no access to an IT system which would enable the collection, analysis, and evaluation of care plans.Organizational learning culture that is responsive to crises (c) [7]. Building resilience also includes the ability to draw conclusions from experience, and from successes and failures, which ensures that actions taken during a crisis situation are effective [19]. International Nurses Day (IND) and the Chartered Institute of Personnel and Development (CIPD) [28] have indicated, among others, the organisational structure and culture and the importance of the ability to introduce changes during a crisis situation as important in building resilience. During the pandemic, the concept of dealing with the crisis was changing due to, among others, new data and feedback from colleagues. A chaotic environment is dynamic and requires dedicating significant attention. It also provides the possibility of making changes which would result in lowering risks and introducing innovative solutions necessary for the correct functioning of the unit.Sound risk adjustment methods include, according to the authors, the Ability to increase capacity to cope with a sudden surge in demand (k) [7]. For the duration of the pandemic, the European Centre for Disease Prevention and Control (ECDC) issued recommendations for establishing three organisational zones within a ward: green, yellow, and red [29]. Recommendations by a team of national consultants established that for one shift (12 h) in a 10-bed unit the staff should constitute: 5 anaesthesia nurses, 2 general nurses, 2 EMTs, and 2 students [30]. The unit was converted to a 2nd reference level unit for 9 patients with COVID-19 by a decision issued by the province governor. The authors included CPD (Continuous Professional Development) as a part of the strategy of alternative and flexible approaches to delivering care (m) [7], which is understood as the provision of cross-training and training necessary to maintain readiness to react to crisis situations (cross-training, that is, delegating employees from other units, in this case for working in the ICU [31]) such as treatment of specific or at-risk population groups, ensuring the provision of services for these groups, and maintaining the quality and safety standards for all services. The training was conducted at the hospital level for hospital admission procedures, isolation procedures, the use of PPE, pathogen transmission routes, handling of infectious material, decontamination of the environment, and handling of medical documentation and patient visits. The deficit of nursing resources, which has been progressing for years, and deficiencies in infrastructure have resulted in care being provided to a limited number of patients. Therefore, a decision was made at the governmental level to open appropriately equipped multi-bed hospitals in sports or exhibition halls. The personnel in these hospitals received additional financial incentives. The basic salary amounted to no less than 200% of the average pay at the position in question [32]. The strategy of Surveillance enabling timely detection of shocks and their impact (e) [7] during the pandemic was based mainly on the distribution of established knowledge, on the creation of procedures intended to, among others, ensure early detection of risks, and on the implementation of actions and quick reactions. The European Observatory [33] emphasises the significant role of the effective flow of information to all participants in a given activity.The ward was qualified as a second reference level unit [34] and provided a high complexity of care (the level of care applies to patients with a life-threatening condition with at least 2 organ dysfunctions, who require pharmacological treatment or mechanical support) for nine intensive care beds, with the possibility of being expanded by two more beds. Actions were undertaken in accordance with the ECDC guidelines [29] and the unit was divided into zones. The green zone contained, among others, the social facilities. The yellow zone, demarcated by a tent placed between the social facilities zone and the medical zone, was intended, among others, for the transport of patients. It was used to decontaminate medical equipment, and the personnel used it to doff PPE after leaving the patients. A simulation of donning and doffing of PPE was conducted [29]; the average time required to don PPE was 5 min [35] with the participation of 2 nurses—the 2nd nurse checked whether the protective suit was airtight, with the complete sets of PPE being prepared by the nurses at the beginning of their shift, in order to reduce the donning time. The use of PPE resulted in discomfort and in the feeling of mental and physical exhaustion, intensified by working in non-air-conditioned rooms (temperature > 25 °C). The red zone was the place where COVID-19 patients were hospitalised. Preparation for providing care to COVID-19 patients included the planning and providing of necessary disposable items. Per one patient, there were 10 infusions operated by infusion pumps and 4 drip infusions. On the average, during one shift (12 h) the nurses prepared approximately 100 continuous infusion syringes and 40 intravenous injections. The time-consuming nature of the procedures required a change in the organisation of work. The shift team (4–5 nurses/shift) was divided into nurses which were (1) preparing and performing doctor’s orders, (2) participating in ward rounds, and (3) providing patient care. After estimating the available competencies of the hospital nurses, attempts were made to obtain competencies necessary at the ICU. Individual talks were made, encouraging cooperation. The ICU ward was supported by surgical department nurses who were providing basic nursing care. It was calculated that the average time necessary to prepare medications and infusions amounted to approximately 30 min/1 patient (drugs administered in the morning) (doctor's orders occur at various hours; most work or changes to the orders occurred after the morning rounds). The time needed for personal care of an intubated patient under mechanical respiration, with a central arterial line and control of diuresis (Foley catheter), is 1 h/2 nurses.IIA health workforce of adequate capacity and with the right skills was connected with the Appropriate level and distribution of human and physical resources (j), which is key for patient safety and is emphasised in nursing organisations’ resilience documents [28,36]. The data indicate that up to 66% of countries have reported insufficient resources for the provision of care during the COVID-19 pandemic [37]. In Poland, the number of nurses employed in the system as of 31 December 2020 amounted to N = 231,612, whereas as of 2021 it was N = 234,117 [38]. There were 5.1 nurses per 1000 inhabitants, and it was one of the lowest values per 1000 patients in the EU [33]. According to the data of the Centre of Postgraduate Education for Nurses and Midwives (CKPPiP) [39], in the year 2020 there were 11,945 specialists in anaesthesiologic and intensive care nursing in Poland. In 2022, this number increased to 14,123 specialists. At this time (23 January 2021), the number of occupied respirator beds exceeded 1500 [33]. The lack of competencies necessary to ensure the optimum quality of care [40], which would prevent avoidable deaths or reduce the cost of care [41], is also of key importance for financial resources. The nurses were frequently the only point of contact between patients and families. They enabled direct contact with patients using IT devices. Thus, the assessment of available nursing resources [42] and a motivated and well-supported workforce (l) 12 to retain the necessary competencies are the priorities which impact the nurses/patients ratio [40]. Despite that fact, the nurses/patients ratio standards were suspended and a 3-month training period for nurses that had not practised the profession for 5 years was introduced, while nurses without the required competencies were requested to obtain them before the date of 31 December 2022 [43]. To meet the needs, the Polish Association of Anaesthesia and Intensive Care Nurses (PTPAiIO) created and implemented emergency disaster plans to support patients and communities while strengthening their workforce, drawing up a programme for rapid training for general nurses [44]. The systemic encouragement of nurses to familiarise themselves with work in the ICU did not achieve the expected effects. Support was provided by medical universities. Medical and nursing students working with patients received academic credits in accordance with the curriculum. The pandemic situation has negatively impacted motivation, causing tensions in relationships and fear against the provision of health services, with a proven relationship to PPE access. This increased anxiety and stress at work during the pandemic [45]. Research conducted by Kilańska et al. before the pandemic in Poland demonstrates the importance of positive practice environment factors. It was established that non-financial incentives, e.g., relationships within the team, support on the part of colleagues (OR = 5), and access to equipment (OR = 2) have a significant impact on the retention of employees and on the decisions of staff to leave their work or their profession, which might happen in a pandemic [46]. The COVID-19 pandemic intensified this phenomenon [45,47]. The impact of staff shortages on the quality of care [48] and on patient deaths [49,50] has been established in many studies, just as the impact of working conditions [51,52].At the level of the studied ward, actions were undertaken which consisted of assessing the nurses/patients ratio, suspending leaves (this applied to 92% of employees at a national level) [53], delegating nurses from other wards, and planning additional medical on-call shifts [6]. (A medical on-call shift is the performance of professional activities by persons referred to in clause 1 outside of normal work hours, at a medical entity which provides inpatient and 24-h health services [54]).Other hospital wards delegated nurses for the provision of nursing care (during the pandemic, specialists and nurses, after a qualification course in Anaesthesiology and Intensive Care, acted as mentors for nursing staff without ICU experience). Adaptation at the ICU was connected to the feeling of insecurity and fear of infection. It should be emphasised that a total of 115,000 healthcare worker deaths from COVID-19 exposure were recorded, with 30% of the infected personnel being nurses [55]. It was reported that their families were also suffering [56], and due to the fear of infecting their loved ones, nurses lived at the wards or isolated themselves (accommodation out of hospital was provided), which frequently resulted in their children remaining without care. Thus, access to personal protective equipment (PPE) became a key issue, and its lack in the initial phase of the pandemic increased the fear of infection. Support was provided by society and by entrepreneurs. However, working in PPE was a hindrance for the nurses, since it impacted their concentration and motor skills, and the temperature in the overalls caused dehydration, cardiac arrhythmias, and headaches [57], hindering the possibility of relieving the physiological needs of the body. Problems in communication with the team and with the patients were also reported, which could pose a risk to patient safety.The low financial resources also had an impact on the lack of personnel [33]. The financial benefits, also for hospital employees, were supposed to provide an incentive. In April of 2020, COVID-19 financial incentives were introduced, to which employees practising a medical profession and providing healthcare services in direct contact with patients with suspected or established SARS-CoV-2 virus infection were entitled. These bonuses were 100% (but no more than PLN 15,000) of gross remuneration (remuneration resulting from an employment contract or civil law agreement of the authorised person in the month for which it is paid out) and also applied to freelance personnel [58]. The rules and the entitled groups were amended by subsequent instructions issued by the Minister for Health (not in the form of a legal act). A total of PLN 8.5 billion (EUR 1,868,747,938.88) was assigned for this purpose [59] in 2021, including (for the first time since the pandemic) PLN 223 million (EUR 49,027,151.81) for one-time financial incentives for non-medical personnel (PLN 5000 per person (approx. EUR 1099)). This bonus was in force until the March of 2022 [60]. The list of entitled persons was drawn up by facility managers. Until today, the manner of assigning of incentives is a cause for employee claims [58]. Until February of 2022, the NFZ fund paid out PLN 8.9 billion (EUR 1,930,585,683.2972) [61] to 105,000 employees in 690 healthcare entities [58].Resilience understood as a health workforce of adequate capacity and with the right skills includes elements (j, l) of the WHO strategy (Table 1). In the described ward, the nurse employment indicator before the pandemic (January 2020) amounted to 2.2 full–time equivalent posts per 1 ICU bed and was in accordance with the standards in force. The number of employed nurses in the ICU *n* = 39, including 3 men. Persons employed for an indefinite duration, *n* = 30, for a definite duration, *n* = 6, and with part-time employment, *n* = 3. The ages of the personnel were between 24 and 65 years. Work experience ranged from 1 year to 37 years, and 54% had more than 25 years of professional experience; a total of 84% of nurses worked in a time equivalent system (extending the daily duration of work to 12 h in a day and night shift); a total of 7 employees were in the basic system (8 h shift) *(working in the basic system constitutes working over a time which does not exceed 7 h 35’ per day and on average 37 h 55’ per week*); a total of 32 nurses—1/3 of personnel—had a BCN (Bachelor of Science in Nursing); a total of *n* = 17 had a MSN (Master of Science in Nursing); and *n* = 7 had medical vocational school training (the old system). It should be emphasised that the absences of personnel in 2020 amounted to 2.3 FTE (full-time equivalent), in 2021—7.9 FTE, and in 2022 as many as 18.4 FTE. The average number of nurses caring for patients during the day vs. the absences of nurses in January of 2020–2022 is shown in Appendix A.The number of patients at the ICU exhibits statistically significant differences at a level of *p* < 0.001. Since we have three versions of the grouping (independent) variable, that is, the year in which detailed calculations were provided, after establishing statistical significance for the model (non-parametric Kruskal–Wallis analysis of variance), we estimated post hoc the statistical significance of the differences between all pairs. An in-depth statistical analysis demonstrated that the only statistically significant differences in the number of patients at the ICU were demonstrated for the year 2020 versus 2021 (*p* = 0.001) and 2021 versus 2022 (*p* = 0.008); between 2020 and 2022 no statistically significant difference in the number of ICU patients was established. A statistically significant relationship exists between the mortality of patients at the ICU and the nursing personnel absences rate (*p* < 0.001). Although during the 2020/2021 period the absences rate increased over 3.4 times, no significant increase in ICU patients’ mortality was observed then. However, comparing 2022 with 2021, the nursing personnel absences rate increased by over 2.3 times, and the ICU patients’ mortality increased approximately twice.The nurses-to-patients ratio in January of every year did not reach the standard level and was the lowest in the second half of January 2022. Therefore, the workload of nurses increased compared with 2020 and 2021. The number of patients hospitalised at the ward in January (before the pandemic) amounted to 24 patients, in other years it was *n* = 20 in 2021 and *n* = 26 in 2022.The number of working hours in January (the number of working hours for healthcare system employees in Poland) of 2020 amounted to 159 h 15′, sickness leaves = 372 h (31 × 12 h) (2.3 FTE), hospitalised patients *n* = 24, and number of deaths *n* = 10/month. A total of 427′ of care should be assigned to a single patient, ×0.66 of the average nurses-to-patients ratio—on average this results in 281.82 (4 h 42′ 22″ of direct care). In 2021 the nurses’ working hours = 136 h 30′; absences = 1080 h (90 × 12 h) = 7.9 FTE; number of patients *n* = 20; per single patient during a day (average nurses to patients ratio—average 0.44 = 427′ = 187.88 (3 h 07′ 52″ × h care); and the number of deaths (month) *n* = 8. In 2022, the nurse working hours = 144 h 05′; absences = 2664 h (222 × 12 h) = 18.4 FTE hospitalised patients *n* = 26; a total of 0.68 nurses were assigned to a single patient during the day, on average 290.36 min (4 h 50′ 36″ × h of care); and the number of deaths = 23 (Figure 1).IIIInformation follows in the system, on the national level, was classified as, among others, monitoring the continuity of essential health services, long-term effects of essential health service disruptions, tracking and addressing the infodemic and health misinformation, collecting or collating data on COVID-19 and comorbidities, and collecting or collating data on post-COVID-19 condition, which due to the lack of IT systems were recorded in Excel and thus were distributed [62]. At a later stage of the pandemic, the eHealth Centre started a system for recording entries into the territory of Poland (EWP) at the central level, financed from EU funds. It monitors the state of epidemic risk in Poland. The data related to incidence of COVID-19, quarantine stays, and isolation stays are collected and analysed for the purposes of pandemic management. EWP contains a central database of all persons under quarantine and in home isolation [63]. Access to medical information was also provided by the Internet Patient Account (IKP) portal which was commissioned before the pandemic, which the patients have used for access to their prescriptions. Since the moment the pandemic started, a record-breaking increase in the number of IKP users was noted, from 875,000 at the beginning of the pandemic to 4,700,000 in January of 2021 [64]. A monitoring and forecasting team was also appointed by the Minister for Health, ensuring the exchange of experiences and the comparison of data of various classes of epidemiological models and of various approaches to the forecasting of the epidemic [65]. Nursing care was not visible in the system. Nurses were not provided with tools for preparing care plans, which were necessary for inclusion in predictive models, even though their importance in this area has been proven [66]. In clinical practice in Poland, nurses do not prepare and do not manage care plans, which may disrupt the care transition process for COVID-19 patients.IVStable funding mechanisms enabling the planning and optimum use of funds in the future guarantee the implementation of multiple elements of resilience. The plan of public expenditure for health care in 2020 amounted to PLN 107.8 billion (EUR 23.7 billion). During this year, it was increased to PLN 123.6 billion, and the actual execution was PLN 116.3 billion (EUR 25.6 billion), including PLN 6.8 billion from the COVID-19 Prevention Fund transferred to NFZ. It amounted to 5.5% of the GDP (GDP—the Gross Domestic Product is a reflection of welfare in a given country; *it is a value of goods and services generated during the year, presented in dollars, Management Encyclopedia)* [67]. These funds in 2020 increased by almost PLN 13.7 billion (EUR 3 billion) compared with 2019 (an increase of 13.3%). In 2021, the plan of public expenditure for health care amounted to PLN 172.9 billion (EUR 37.5 billion) [59] and amounted to 6.6% of GDP and increased by 21 billion compared with 2019. In 2022, the plan amounted to PLN 130,590,664 (EUR 27,263,186,638) [68] and was increased in the middle of the year by PLN 6.5 billion/(EUR 1.3 billion) [69]. At the national level, agreements with private health facilities to deliver essential health services supported through public funds were deemed important. Such agreements were made in 46% of countries worldwide [70]. The ICU services were financed by an agreement with the public payer (National Health Fund—NFZ). Units of the 1st and 2nd level of COVID-19 protection (traditional general hospitals) (in the pandemic period, hospitals were divided into traditional hospitals, that is the ones which before the outbreak were providing health care services in a closed system and into newly opened temporary institutions in public buildings, such as, e.g., a sports hall) received funds for keeping beds in readiness [71], was amounting to, on an average EUR 157 (PLN 717) [72] per day/per bed (compensation for the lost income from health services provision agreement) [71]. The National Health Fund also financed beds not covered by an agreement at EUR 21.96 (PLN 100)/day/bed and access to respirators not covered by an agreement at EUR 43.9 (PLN 200)/day/1 respirator. The pandemic resulted in limitation of UHC (cancer screening diagnostics were limited and waiting times for specialists increased, among others) and shortening of the average lifespan of Poles. The healthcare debt is increasing [33].Among the variables on which the preparation of the system to react to crises depends, the following were listed: Stable funding mechanisms (V) enabling the planning and optimum use of funds in the future. In the case of the fees for the treatment of a patient on mechanical ventilation at the Intensive Care Unit, the payer provided from PLN 978 (approx. EUR 215) to PLN 5691 (approx. EUR 1250)/person/day (PD). This amount depended on the point rate assumed at the moment a contract was signed with the NFZ. Additionally, respirator beds outside of the ICU were also financed, at the amount of PLN 1154 (the amount *is a result of the multiplication of the product value equal to PLN 1.16*) (approx. EUR 253)/person/day [72]. The financing of healthcare employees’ remuneration was specified by an Act of Parliament [59]. Differences in the financing of patient treatment resulted from the use of varied equipment and medical procedures, pursuant to the regulation by the Polish Minister for Health. The hospital has received, among other things, PPE from the Material Reserves Agency, a government institution intended to secure strategic reserves for crisis situations [73]. The hospital also received financing from EU Funds for the prevention and control of SARS-CoV-2 virus infections and the spread of the disease caused by this virus in humans within the Lodz Province.VAdequate costing of health services; comprehensive health coverage (i): The patients without the right to universal health coverage during the pandemic were granted the right to use the available health care services. This also applied to foreigners. The services provided as a result of controlling the pandemic, treating the disease, and the related sanitary and epidemiological tests were financed from the state budget [74].

## 4. Discussion

This study aimed to assess the healthcare system during the COVID-19 pandemic, both at the local and central levels. The complexity of the processes of treatment and care of ICU patients became a significant challenge for the ward’s managers. It is precisely during the pandemic that the ICU became a focus for the issue of the healthcare system’s resilience. The presented case study concerned the assessment of the six components indicated by the EC. They include good governance, sound risk adjustment methods, and also health workforce of adequate capacity and with the right skills. The next important elements are information follows in the system, stable funding mechanism, and the last one, adequate costing of health services [8]. Despite a significant body of work on resilience at the ICU, few studies have been conducted concerning the resilience understood as preparation, adaptation, or the ability to react and draw conclusions from the occurring crisis situation. The available literature mostly refers to mental endurance in the ICU. Despite a small group examined by the study, the results reflect the observations of other authors. The study involved resources, effective communication, managerial skills, and management in a crisis situation of the COVID-19 pandemic. The results concerning resources have indicated unequivocally that RNs are the most important element of crisis management, which is important in maintaining the continuity of ICU operations [75,76,77]. Even though resilience applies not only to the nursing staff, during the pandemic the staff were of key importance in responding to the needs of the system and in identifying risk in patients. This article focuses on nurses, since evidence suggests their significant impact on extending hospitalisations, increase in mortality, and increase in nosocomial infections among ICU patients. Negative effects of a hospital stay may be related to insufficient nursing care [78]. The authors also emphasise the significant impact of the nurses-to-patients ratio on the quality of care and on the associated risk of complications related to hospitalisation [77,79]. The material resources and access to PPE were also important during this time [80,81,82], as well as the ability to conduct simulations of technical skills [83], which rallied the nurses, providing them with psychological safety [76]. These results are also consistent with other studies concerning the importance of PPE availability and the risk of an increase in COVID-19 incidence among the employees and their families [1,80]. This could be an additional cause of professional absences and of the inability to provide continuity of nursing care on an appropriate level. Our study demonstrates that the preparation or adaptation of the infrastructure to the existing needs is important. In a crisis situation, the establishing of safety zones also acted as a prevention measure for the ICU nurses. This is confirmed by both ECDC recommendations [29] and also by research [84]. A well-adapted infrastructure with established zones allows for resting safely without increasing the risk of a COVID–19 infection. This also impacts the ICU nurses’ feeling of safety.

During the pandemic, nurses managing the ICU had to face multiple challenges. During the first wave there were problems with lack of information about the exact paths of transmission, with insufficient availability of PPE [85] and with pandemic disinformation [86]. The organisation of work and resources was frequently related to the ability to make rapid decisions under a time pressure. According to the surveyed persons, the most important managerial skills were effective decision making, skilled motivation, and effective communication. The lack of the aforementioned skills could lead to ethical conflicts [87,88]. It should be noted that a manager’s resilience translates to the team’s resilience, as described by Liang [89]. Frequently, nurse managers were required to manage a team which simultaneously suffered from work overload and from fear for their health [82,89,90]. This is why mental resilience of the leaders was priceless. Amongst the informational, decisional, and interpersonal roles, the surveyed personnel indicated the continuous assessment of the situation, composure when making decisions, the ability to delegate tasks, and possession of knowledge about the situation that has occurred as the most important [66,91]. Providing support to employees, sharing of knowledge, and motivating employees has a significant impact on their resilience [90] and on the ICU nurses’ resilience in connection with a positive practice environment [91]. The ability to participate in workshops, regular assessment of management quality, and spreading the workload evenly were all important during the pandemic.

Good communication is one of the primary working tools, and effective communication forms the basis of management. We use the provided information to plan, organise, or to motivate employees to action. An important function of information is the integration of jointly undertaken tasks and the exchange of experiences [92]. The quality of information impacts work satisfaction and the quality of cooperation within the team. This was also reflected in the research, where the survey expected full information to be provided to all employees. The AACN (American Association of Critical-Care Nurses) in its guidelines emphasises the importance of full information sharing between the medical personnel, patients, and their families [93]. During the pandemic, the forms of communication between the personnel, and also between the patients and the personnel, have changed significantly due to the barrier formed by the PPE [94]. According to the surveyed persons, electronic transmission of information was an important part of communication in a crisis situation. This form of communication, recommended by the WHO, reduced the risk of falling ill with COVID-19 [95,96].

Risk management also includes access to resources and the ability to anticipate risk. In the case of a pandemic, unusual phenomena have to be taken into account, such as the rapid transmission of a virus. Due to the scale of the threat and the key importance of healthcare employees in ensuring health safety during the pandemic, procedures for prevention of SARS-CoV-2 infections were introduced. The procedures established in the recommendations included, among others, the manner of handling of patients suspected of a coronavirus infection, the use of personal protective equipment, and the decontamination of medical equipment after coming into contact with the virus [96,97]. The ward preparation phase included the assessment of current human, physical, informational, and financial resources (including the PPE described earlier). The risk assessment should apply to the most important elements of each resource, which was reflected in the research results, where the surveyed persons have indicated the possibility of assessing SARS-CoV-19 infection risk for the entire personnel working at the ICU to be important. Moreover, the Lancet Commission [85] in its document emphasised the significance of effective communication of infection risk, of the provision of PPE, of training programmes, and of provision of adequate staffing.

An important element of resilience includes ensuring appropriate financing of the health care system, which should translate into, among others, provision of adequate remuneration [97] and the possibility of making urgent purchases during a crisis situation [19]. This will have an impact on the mobilisation of the personnel to action, and therefore will have a significant impact on maintaining the stable operation of the system. This was also confirmed by the surveyed persons. The HCS financing should be provided by the government, but also by private founders, which occurred during the pandemic, where private decision makers (business owners) provided PPE to HCS employees [19]. Although in our study we did not ask directly about this element, some question descriptors were qualified as its components.

The notion of adequate costing of health services takes on a particular meaning during the pandemic. Understanding of processes, which during this period are ambiguous and complex is of key importance [97]. The EC emphasises the importance of the assessment of the effectiveness of undertaken actions concerning, among others, the distribution of resources using accurate cost calculations and the establishment of key tasks or investments [8]. The pandemic demonstrated the weak points of the system and the need for continuous assessment of the performance of the assumptions prioritised at the time. This was related, among others, to the reorganisation of many health care facilities into COVID-19 hospitals, thus causing/increasing the health debt.

## 5. Limitations

The main limitation of this report is the description of the situation on the basis of one ICU. Attempts to describe other ICUs have not been successful due to the COVID-19 period and the workload at this time. Another limitation was access to structured data that would support the description of care effectiveness and description of part VI Resilience according to the EU. The description of financial human resources was possible only on the basis of publicly available data. Part of human resources has been described only in the example of an ICU, which also shows the international context.

## 6. Implications for Practice

The perspective of health care system resilience applies to organisations and institutions, but primarily to the smallest units of hospital wards. The ability to manage a pandemic crisis situation has a significant impact on the quality of care, but also on the general efficiency of the health care system. The initial assessment of the ward’s ability to handle a crisis situation has prompted the following recommendations for nursing practice in the ICU:

Good governance means the attention paid to maintaining good communication within the team and the ability to manage in a situation of continuous changes. An inseparable element of this process is the obtaining of knowledge and the ability to transfer the knowledge, ensuring the continuity of resources.

Sound risk adjustment concerns following the procedures, the possibility of detecting threats in advance, training concerning the transmission paths and prevention of infections among nurses, and also learning lessons from failures.

A Health workforce of adequate capacity and with the right skills means the assessment of the available human resources, the possibility of professional adaptation, and providing support to the personnel. An important element is the financial support and continuity of material resources (including PPE) which facilitate work.

Information follows in the ICU applies to all employees of the ward. The most important element is the transfer of proven knowledge, prevention of disinformation, and regular feedback on every level of management.

## 7. Conclusions

1. A case study analysis referring to the EC/WHO Resilience Model demonstrated that human resources are a very sensitive element of the organisational-level resilience-building process. It was especially visible during the period of 'care burnout' of the personnel and of the increased absences.

2. Lowering the standards of care due to an increase in the absences of qualified nurses and of the number of patients provided care by a single specialist RN resulted in an increased number of patient deaths.

3. The pandemic has demonstrated weak areas of resilience, such as communication, lack of procedures available at the unit level, and difficulties with ensuring necessary competencies for the patient care. Procedures adequate to the resilience model should be implemented at the unit level and they should be subjected to accreditation during hospital assessment.

## Figures and Tables

**Figure 1 medicina-59-00946-f001:**
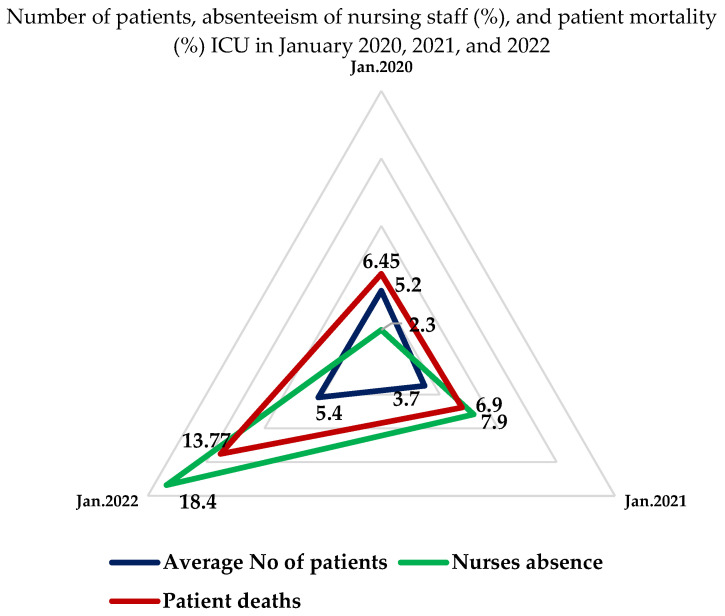
A list of data for the month of January in the years 2020, 2021, and 2022 in Pirogow Hospital in Lodz, Poland.

**Table 1 medicina-59-00946-t001:** ICU resilience matrix with the referred to I–VI European Commission [8] elements of resilience and the 13 strategies of the WHO resilience classification [7].

I. Good Governance	II. Sound Risk Adjustment Methods	III. A health Workforce of Adequate Capacity and with the Right Skills	IV. Information Flows in the System	V. Stable Funding Mechanisms	VI. Adequate Costing of Health Services
(a) Effective and participatory leadership with a strong vision and communication.	(e) Surveillance enabling timely detection of shocks and their impact.	(j) Appropriate level and distribution of human and physical resources.	(d) Effective information systems and flows.	(f) Ensuring sufficient monetary resources in the system and flexibility to reallocate and inject extra funds.	(i) Comprehensive health coverage.
(b) Coordination of activities across government and key stakeholders.	(h) Purchasing flexibility and reallocation of funding to meet changing needs.	(l) Motivated and well-supported workforce.		(g) Ensuring stability of health system funding through countercyclical health financing mechanisms and reserves.	
(c) Organizational learning culture that is responsive to crises.	(m) Alternative and flexible approaches to delivering care.				
	(k) Ability to increase capacity to cope with a sudden surge in demand.				

Legend: I–VI, the elements of resilience according to the European Commission [8]; (a)–(m), the elements of resilience according to the WHO document.

## Data Availability

Non-digital data supporting this study are curated by Dorota Kilańska.

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
