# Peer review of "The Effectiveness of Healthcare System Resilience during the COVID-19 Pandemic: A Case Study"

_medicina, 2023, doi:10.3390/medicina59050946_

Round 1
Reviewer 1 Report
The study has significant methodological weaknesses. Cannot understand the data collection strategies used. Documentary analysis? How was the comparison with the pillars carried out? was it observation? It is recommended to use a best practice guide such as STROBE (for example). Identify the host institution, did you have permission from the ethics committee to conduct the study?
What were the criteria for the selection of the panel of experts? what were the inclusion criteria?
What questions did the questionnaire have? how reliable was it?
What is the relationship between the mixed methodology used and how the results are presented?
Author Response
Dear Reviewer!
Thank you very much for the detailed analysis of the publication. All insights are very valuable to us and helped us to improve the quality of the prepared text. Thank you very much for your favorable view of the material and valuable methodological tips.

Reviewer 2 Report
Very interesting topic, the researchers did a great job to show the preparations at an ICU in Poland intended to ensure resilience, and to show the methods of reacting during the COVID–19 pandemic on both central and local level. However, the article could be strengthened through:
Introduce every acronym before using it in the text. The first time you use the term, put the acronym in parentheses after the full term, and COVID 19 in line 39, and UHC in line 46, ICU in line 64.
Table 1 could be moved to the appendix or re-designed.
Consistency in citation, like using footnotes in Pages 5-8. Also, please check page numbers, it is not in order.
Please provide more details in method section.
Add more towards scope of the problem in introduction section.
Good luck
Author Response
Review 2
Dear Reviewer!
Thank you very much for the detailed analysis of the publication. All insights are very valuable to us and helped us to improve the quality of the prepared text. Thank you very much for your favorable view of the material and valuable methodological tips.
|
1. Introduce every acronym before using it in the text. The first time you use the term, put the acronym in parentheses after the full term, and COVID 19 in line 39, and UHC in line 46, ICU in line 64. |
Thank you very much for the suggestion, we have corrected it in the text. |
|
2. Table 1 could be moved to the appendix or re-designed. |
Thank you very much for your suggestions. Table 1 has been moved to the results section, the method of assigning variables is described in the methodology. |
|
3. Consistency in citation, like using footnotes in Pages 5-8. Also, please check page numbers, it is not in order. |
We have analyzed the footnotes and corrected them as suggested. |
|
4. Please provide more details in method section. |
The methodology was supplemented with all the elements that were implemented during the study |
|
5. Add more towards scope of the problem in introduction section. |
We supplemented the introduction by introducing a broader description of the problem, and moved the table to the results |
Once again we would like to sincerely thank you for a very comprehensive, insightful and in many places accurate review. All of the above comments are highly valuable for the comprehensiveness of the paper and our own scientific development.
Sincerely yours,
Authors

Reviewer 3 Report
Generally well written, easily readable and an interesting topic.
Introduction:
- Abbreviations such as UHC are used before they are defined (first used line 46, defined line 81). ICU is a well known abbreviation but should be defined on first use.
- COVID should be COVID-19, there are multiple times it has been shortened to just COVID (line 17, 39 etc etc).
- The introduction and throughout the article uses the term resilience on it's own a lot, however the term actually has a much wider definition than what is being discussed in the introduction. It needs to be put into context as either (healthcare) system resilience or resilient (healthcare) systems. At present the language is quite confusing as patient resilience and system resilience are both potential interpretations.
- on line 85 table 1 is abbreviated as Tab. 1, whereas on line 92 it is referred to as table 1. Ans also Table 1 on line 96.
- note it is difficult to keep track of the references [1],[2] etc, the Table 1 items (1),(2), etc and the Table 1 domains (I),(IV) etc in the text, especially in the paragraph (lines 71-75). At the very least I would recommend using (a),(b), etc instead of (1),(2), etc. This also flows onto the further sections.
- I note that the Table 1 matrix was developed in the first section of the results which therefore makes no sense why it is included in the introduction and is discussed in the introduction.
Materials and methods:
The methods have very limited detail and does not specify how the qualitative analyses undertaken for the two "case-studies" or any detail on the analyses. The quantitative survey also has very few details on the questionnaire development, participants or analysis.
Results:
- There is reference to analysis of literature and semantic analysis to create Table 1 but limited detail on exactly how this was carried out. In addition this process is not referenced in the methods section.
- note missing reference on line 117.
- There is a lot of information that relate to national Poland ICU healthcare and/or EU recommendations lines 105-175 which doesn't appear to directly correspond to the methodology of case studies of a single hospital. IT is important contextual information but not specified in the methods.
- Lines 176-204 (in a slightly different format as well) appears to relate to the case study information, but appears to be completely disconnected from the previous paragraph.
- there appears to be odds rations (OR) reported on line 237. It is not clear where they have originated from, I assume they are related to the survey referenced in the methods (but lacking any detail) as the two closest references are not the source of the ORs.
- The section on "III A health workforce of adequate capacity and with the right skills" of the results section appears to be a mix of national and single hospital results. This does appear to be were the survey and case studies are primarily focused (but not clear from the writing).
- Figures 1 and 2 should be labelled with what I assume will be the name of the hospital under examination.
- I note that Figure 1 is labelled as Fig. 1 and Figure 2 as Figure 2. Need to be consistent in labeling.
Conclusions etc:
It is hard to comment on these sections without the detail about the methodologies. It is an interesting read.
More comments:
1. What does it add to the subject area compared with other published material? It potentially adds a slightly different slant to the COVID-19 narrative by bringing in the healthcare system resilience.
2. What specific improvements should the authors consider regarding the methodology?
What further controls should be considered? There is a complete lack of detail for the methodology and it is not clear which component of the methodology is responsible for which results. Also some components of the results are not addressed in the materials and methods section. There appears to be some operational data both locally and nationally as well as the development of Table 1 referenced in the results but not directly referenced in the methodology.
3. Without a clear methodology, I am unsure on the source of the data for the figures 1 and 2, it is either hospital operational data or results from the survey briefly referred to in the methodology (but lacking any details about participants or content of the survey). Table 1 is a fundamental part of the research, however as it was developed at the beginning at the results section, place the table and discussion about the table in the introduction appears to be problematic.
Author Response
Review 3
Dear Reviewer!
Thank you very much for the detailed analysis of the publication. All insights are very valuable to us and helped us to improve the quality of the prepared text. Thank you very much for your favorable view of the material and valuable methodological tips.
Introduction:
|
1. Abbreviations such as UHC are used before they are defined (first used line 46, defined line 81). ICU is a well known abbreviation but should be defined on first use. |
Thank you very much for the suggestion, we have corrected it in the text. |
|
2. COVID should be COVID-19, there are multiple times it has been shortened to just COVID (line 17, 39 etc etc). |
Thank you very much for the suggestion, we have corrected it in the text. |
|
3. The introduction and throughout the article uses the term resilience on it's own a lot, however the term actually has a much wider definition than what is being discussed in the introduction. It needs to be put into context as either (healthcare) system resilience or resilient (healthcare) systems. At present the language is quite confusing as patient resilience and system resilience are both potential interpretations. |
Thank you very much for the tips, we have analyzed the literature and supplemented it as suggested. |
|
4. on line 85 table 1 is abbreviated as Tab. 1, whereas on line 92 it is referred to as table 1. Ans also Table 1 on line 96. |
We have corrected as suggested. |
|
5. note it is difficult to keep track of the references [1],[2] etc, the Table 1 items (1),(2), etc and the Table 1 domains (I),(IV) etc in the text, especially in the paragraph (lines 71-75). At the very least I would recommend using (a),(b), etc instead of (1),(2), etc. This also flows onto the further sections. |
We have corrected as suggested |
|
6. I note that the Table 1 matrix was developed in the first section of the results which therefore makes no sense why it is included in the introduction and is discussed in the introduction. |
Table 1 has been moved to the results section, the method of assigning variables is described in the methodology |
|
Materials and methods: |
|
|
7. The methods have very limited detail and does not specify how the qualitative analyses undertaken for the two "case-studies" or any detail on the analyses. The quantitative survey also has very few details on the questionnaire development, participants or analysis. |
The methodology was supplemented with all the elements that were implemented during the study. The article has been divided into a quantitative and qualitative part. In the version submitted for review, elements concerning quantitative analysis have been preserved. We have removed them from the publication. We are very sorry that we missed this part |
|
Results: |
|
|
8. There is reference to analysis of literature and semantic analysis to create Table 1 but limited detail on exactly how this was carried out. In addition this process is not referenced in the methods section. |
We supplemented the description of the creation of table 1 in the methodology, while the table itself was moved to the results. |
|
9. note missing reference on line 117. |
Thank you very much for pointing out the deficiencies in the text, they have been supplemented. |
|
10. There is a lot of information that relate to national Poland ICU healthcare and/or EU recommendations lines 105-175 which doesn't appear to directly correspond to the methodology of case studies of a single hospital. IT is important contextual information but not specified in the methods. |
We have improved the description of the methodology by referring to the research questions posed during the design of the study, perhaps in this version it will be clearer and finding a reference will facilitate the analysis of the material. |
|
11. Lines 176-204 (in a slightly different format as well) appears to relate to the case study information, but appears to be completely disconnected from the previous paragraph. |
The text has been reformatted according to the guidelines. |
|
12. there appears to be odds rations (OR) reported on line 237. It is not clear where they have originated from, I assume they are related to the survey referenced in the methods (but lacking any detail) as the two closest references are not the source of the ORs. |
The result concerns a study conducted in the work environment of nurses in Poland before the pandemic, they showed a significant impact of the variables that relate to the resilience model described in the case study, the footnote regarding the study is in the text of the article - line no 349 Ref. no. 45 |
|
13. The section on "III A health workforce of adequate capacity and with the right skills" of the results section appears to be a mix of national and single hospital results. This does appear to be were the survey and case studies are primarily focused (but not clear from the writing). |
Hospital data is located against the background of national data, but the research data was data from a single hospital. We tried to show the national context. |
|
14. Figures 1 and 2 should be labelled with what I assume will be the name of the hospital under examination. |
We corrected in the text. Thank You very much. |
|
15. I note that Figure 1 is labelled as Fig. 1 and Figure 2 as Figure 2. Need to be consistent in labeling. |
We corrected in the text. |
|
Conclusions etc: |
|
|
16. It is hard to comment on these sections without the detail about the methodologies. It is an interesting read. |
We analyzed the conclusions in detail and in the current text we referred them to the research problems described in the methodology. Thank you very much for your suggestions, which made us rethink the conclusions. |
|
More comments: |
|
|
17. What does it add to the subject area compared with other published material? It potentially adds a slightly different slant to the COVID-19 narrative by bringing in the healthcare system resilience. |
We tried to show in the conclusions what should be changed. |
|
18. What specific improvements should the athors consider regarding the methodology? |
The meodology has been rewritten, specifying more of the elements that were designed during the study. |
|
19. What further controls should be considered? There is a complete lack of detail for the methodology and it is not clear which component of the methodology is responsible for which results. Also some components of the results are not addressed in the materials and methods section. There appears to be some operational data both locally and nationally as well as the development of Table 1 referenced in the results but not directly referenced in the methodology. |
We have arranged the methodology to maintain consistency and show which Conclusions come from which research problem. |
|
20. Without a clear methodology, I am unsure on the source of the data for the figures 1 and 2, it is either hospital operational data or results from the survey briefly referred to in the methodology (but lacking any details about participants or content of the survey). Table 1 is a fundamental part of the research, however as it was developed at the beginning at the results section, place the table and discussion about the table in the introduction appears to be problematic. |
We improved the methodology and described in more detail the manner in which tab. 1; we hope it is more transparent and will support the review process. Our deepest apologies for not getting it right the first time.
|
Once again we would like to sincerely thank you for a very comprehensive, insightful and in many places accurate review. All of the above comments are highly valuable for the comprehensiveness of the paper and our own scientific development.
Sincerely yours,
Authors

Round 2
Reviewer 1 Report
Your article has had a substantial improvement. However, the objective described in the abstract should be the same as the one placed in the body of the paper.
Author Response
Dear Reviewer!
Thank you very much for your support in improving the quality of the material for publication. We have made possible fixes, which we write about below
|
1. However, the objective described in the abstract should be the same as the one placed in the body of the paper. |
Thank you very much for finding this unforgivable error. We have supplemented the abstract in accordance with the content in the text. |
Once again we would like to sincerely thank you for a very comprehensive, insightful and in many places accurate review.
Sincerely yours,
Authors

Reviewer 2 Report
Well done
Author Response
Thank you very much for your support in improving the quality of the published material.
Authors
Reviewer 3 Report
Some minor tweaks:
- The section heading "3. Results" appears to have been accidently deleted.
- figure 1 in the appendix, I have just noticed that the legend for sickness/absence that the years are out of order, also that the RN/Patient ratios for 2020 and 2022 have similar colors for someone like myself who is red/green color blind but also for purposes of printing the figure in grey scale may need some more distinguishing patterns.
Author Response
Dear Reviewer!
Thank you very much for your support in improving the quality of the material for publication. We have made possible fixes, which we write about below.
Introduction:
|
The section heading "3. Results" appears to have been accidently deleted. |
Thank you very much for the suggestion, we have corrected it in the text. |
|
Figure 1 in the appendix, I have just noticed that the legend for sickness/absence that the years are out of order, also that the RN/Patient ratios for 2020 and 2022 have similar colors for someone like myself who is red/green color blind but also for purposes of printing the figure in grey scale may need some more distinguishing patterns. |
We improved the readability of the chart. Unfortunately, the order in the legend cannot be changed. This is an automatic option, we will ask editors to assist us in the printing process, where more advanced options may be available. |
Once again we would like to sincerely thank you for a very comprehensive, insightful and in many places accurate review. All of the above comments are highly valuable for the comprehensiveness of the paper and our own scientific development.
Sincerely yours,
Authors
